# Improved Fixed-Frequency SOGI Based Single-Phase PLL

Djordje Stojic [1,*], Tomasz Tarczewski [2], Lukasz J. Niewiara [2] and Lech M. Grzesiak [3]

1 Electrical Institute Nikola Tesla, University of Belgrade, 11000 Belgrade, Serbia
2 Institute of Engineering and Technology, Nicolaus Copernicus University in Torun, 87-100 Torun, Poland
3 Institute of Industrial Electronics, Warsaw University of Technology, 00-661 Warsaw, Poland
* Correspondence: djordje.stojic@ieent.org

**Abstract:** In this paper, an improved single-phase second order generalized integrator (SOGI) fixed-frequency phase-locked loop (FFPLL) is presented. The proposed improvement comprises the modification of the PLL input signal estimated phase angle correction factor, which is in this paper calculated and implemented with the exactly accurate value, while in the existing literature the approximated correction value is employed. Also, in this paper, the FFPLL with DC offset is presented, together with the corresponding estimated angle correction technique. Furthermore, the PLL with the positive sequence separation is outlined, based on the new FFPLL structure. The proposed technique is analyzed and verified by simulation and experimental runs, which proved the accuracy and efficiency of the proposed PLL technique. Furthermore, a corresponding PLL parameter values tuning procedure is presented that illustrates the dynamic performance improvements that SOGI based FFPLL introduces when compared with SOGI based PLL. Consequently, FFPLL combined with the proposed new estimated angle correction factor represents a significant improvement when compared to the conventional SOGI based PLL.

**Keywords:** estimated angle correction; DC offset compensation; fixed-frequency orthogonal signal generator; positive sequence separation; single-phase PLL; SOGI

## 1. Introduction

The single-phase PLL represents a significant task in many engineering applications, including various types of grid-connected single-phase power converters. Namely, accurate and fast frequency and phase angle estimation of the grid voltage is required, which needs to operate with the PLL input signals contaminated by higher harmonics, voltage dips, and also frequency and phase angle variations. Consequently, several comprehensive single-phase PLL survey papers have been published [1–3], in which major PLL design problems and issues are reviewed and presented.

### 1.1. Motivation

The main motivation behind the work outlined in this paper emerges from the possibility to improve significantly the estimation precision of existing FFPLL solutions. Also, the importance of the FFPLL solutions is based on several analyses, which show that implementation of frequency non-adaptive FFPLL solutions, when compared to conventionally used frequency adaptive PLL solutions, introduces significant improvements in resulting stability, maximum response speed, and phase-locked loop robustness of operation.

Namely, the main drawback of existing FFPLL solutions comprises the approximated estimated phase angle compensation, contrary to the proposed solution, which is based on the analytically accurate compensation factor.

### 1.2. Literature Review

Generally, the single-phase PLL solutions can be divided into two main groups–power PLL [3] and PLL based on different orthogonal signal generators (OSG) [1]. Power PLL

algorithms represent simple and effective solutions, which, however, suffer from the significant double main frequency component, which results in the substantially reduced PLL response speeds that need to be tuned. Orthogonal signal generator based solutions, however, result in much higher response speed, and they can operate with a DC offset present at the PLL input.

There is a wide range of different OSG filters and techniques [1] proposed in the literature, which is outside the scope of this paper. However, one of the most commonly used OSG algorithms, SOGI [4], represents the basis of the FFPLL solution proposed in this paper. Namely, SOGI based applications are commonly used as adaptive resonant frequency filters fit for the OSG, which can also successfully be applied in the case when a DC offset is present at the PLL input [4,5]. However, the fact that the single-phase PLL closed-loop algorithm operates with an adaptive frequency OSG filter results in a non-linear PLL operation, which introduces difficulties in parameter tuning in order to enable stable and fast PLL operation.

Consequently, in order to avoid the nonlinear adaptive frequency SOGI filter application, single-phase PLLs are proposed using OSG with a fixed frequency tuned SOGI [6–10]. In [9], an OSG is implemented based on the fixed-frequency SOGI, with the accurate orthogonal voltage amplitude and phase angle corrections, which are necessary because of the estimation error caused by the fixed frequency SOGI tuning. However, in [9] a complex input signal frequency estimation method is proposed, when compared to other different FFPLL solutions.

In [8], a detailed analysis of the FFPLL structure and dynamics is presented, while in [7] the original FFPLL structure is outlined. Also, in [8] a modification of the basic FFPLL [7] structure is proposed, with the increased PLL frequency and phase angle estimation speed. In [10], the FFPLL based estimator is presented, used to separate the positive and negative sequence components in the non-symmetrical PLL input signals.

Regarding the application of the FFPLL algorithms in radio frequency (RF) and microwave applications, it is limited by the features of the employed control platform. Namely, in order to implement an FFPLL based on a digital signal processing (DSP) in an RF application, a specialized RFSoC platform [11] could be employed. However, there is a possibility of an analogue FFPLL application [12], which can overcome shortcomings of a DSP based solution.

### 1.3. Contribution and Paper Organization

In this paper, the modification of the original FFPLL [7] structure is proposed, with the accurately calculated phase angle estimate correction value, as opposed to [7] in which an approximation is employed. Namely, in conventional FFPLL applications [7] phase compensation value is approximated for the estimated input signal frequency value close to fixed resonant frequency of the employed SOGI term, while the new proposed solution comprises an analytically calculated accurate phase compensation factor. In this way, accurate operation of the FFPLL for the much wider differences between the input signal frequency and SOGI term fixed resonant frequency value, which was not the case in conventional FFPLL solutions. Also, in this paper, the FFPLL structure is analyzed to operate with a DC offset present at the PLL input together with the PLL based on the input signal positive sequence separation, which is not the case in any of the existing FFPLL structures.

This paper comprises six sections. In Section 2, the existing FFPLL structures are outlined and compared. In Section 3, the improved FFPLL structure is proposed, including the modification, which enables FFPLL operation with the DC offset at the PLL input. In Section 4 the simulation results are presented, while in Section 5 the experimental tests are outlined.

Consequently, the problem statement comprises an effort to improve the phase angle estimation accuracy in the complete input signal frequency range for the existing FFPLL

solutions, which are based on several analyses [7,8] dynamically superior to conventionally used frequency adaptive SOGI based PLLs.

## 2. Fixed-Frequency PLL

In this section, the existing FFPPL solutions are presented and analyzed. Namely, the FFPLL is derived from the single-phase PLL with the adaptive frequency SOGI filter used for the orthogonal signal generation, which is outlined in Figure 1.

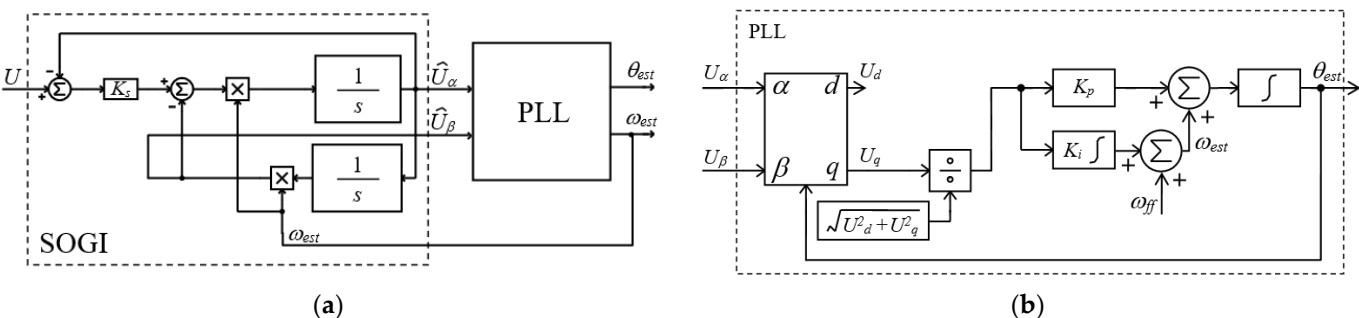

(**a**)　　　　　　　　　　　　　　　　　　　　　　　　　　　(**b**)

**Figure 1.** A conventional adaptive frequency SOGI based single-phase PLL, (**a**) general structure with SOGI input section, and (**b**) PLL section.

In Figure 1 $U$ represents the PLL input, $\hat{U}_\alpha$ and $\hat{U}_\beta$ orthogonal components generated by SOGI, $K_s$ the SOGI parameter, $\omega_{est}$ and $\theta_{est}$ the estimated input signal frequency and phase angle, $U_\alpha$, $U_\beta$, $U_d$, and $U_q$ the PLL input and auxiliary signals, $K_p$ and $K_i$ the PLL parameters, and $\omega_{ff}$ the estimated frequency feed-forward value. The SOGI parameter value $K_s = 2$ is commonly used, while the PLL parameters $K_p$ and $K_i$ are commonly designed by the symmetrical optimum technique [2].

However, in [8] a shortcoming of the adaptive frequency PLL in Figure 1 is analyzed, caused by the nonlinear SOGI filter operation, which restricts the resulting PLL dynamics. Consequently, the FFPLL is proposed [7,8] in which the SOGI filter operates with a fixed frequency, resulting in linear orthogonal signal generation. Namely, the FFPLL is designed based on the following SOGI Equations (1) and (2), derived from the structure outlined in Figure 1a. Namely, Equations (1) and (2) represent the basis for the derivation of the FFPLL operating equations, which are outlined in the following part of the paper.

$$\hat{U}_\alpha(s) = \frac{K_s s \omega_{est}}{s^2 + K_s s \omega_{est} + \omega_{est}^2} U(s) \tag{1}$$

$$\hat{U}_\beta(s) = \frac{K_s \omega_{est}^2}{s^2 + K_s s \omega_{est} + \omega_{est}^2} U(s) \tag{2}$$

As it was shown in [1], for $s = j\omega_{est}$ $\hat{U}_\alpha(j\omega_{est})$ is equal to $\hat{U}(j\omega_{est})$, while $\hat{U}_\beta(j\omega_{est})$ is orthogonal with $\hat{U}(j\omega_{est})$. However, for the input signal frequency $\omega \neq \omega_{est}$ this is not the case, which is of special interest for the FFPLL applications.

Namely, in order to avoid the PLL operation with the adaptive SOGI filter, the SOGI rated frequency is fixed to the reference value $\omega_n$ (usually equal to $2\pi50$ rad/s), with the corresponding PLL structure outlined in Figure 2, which includes the fixed-frequency SOGI filter (FFSOGI) [7,8].

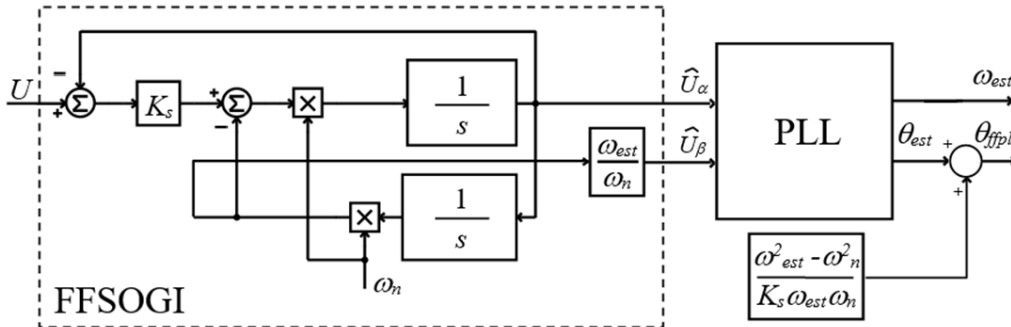

**Figure 2.** The basic structure of FFPLL1.

Figure 2 presents the FFPLL1 structure, where $\omega_n$ represents the rated SOGI frequency, while $\theta_{ffpll}$ represents the resulting estimated phase angle value. The main features of the FFPLL1 are outlined by analyzing the following Equations (3) and (4) of the FFSOGI output signals in Figure 2.

$$G_\alpha(s) = \frac{\hat{U}_\alpha(s)}{U(s)} = \frac{K_s s \omega_n}{s^2 + K_s s \omega_n + \omega_n^2} \tag{3}$$

$$G_\beta(s) = \frac{\hat{U}_\beta(s)}{U(s)} = \frac{K_s \omega_n \omega_{est}}{s^2 + K_s s \omega_n + \omega_n^2} \tag{4}$$

In (3), $G_\alpha(s)$ represents the transfer function from the PLL input to the $\alpha$ axis output, while $G_\beta(s)$ represents the transfer function from the PLL input to the $\beta$ axes output. By analyzing (3) and (4), it can be concluded that for $s = j\omega_{est} \hat{U}_\alpha(j\omega_{est})$ and $\hat{U}_\beta(j\omega_{est})$ are orthogonal, which enables the PLL to estimate successfully the PLL input signal frequency. However, there is a phase angle error $\Delta\varphi$ (5) between $\hat{U}_\alpha(j\omega_{est})$ and $\hat{U}(j\omega_{est})$, which results in the erroneous PLL output phase angle estimate $\theta_{est}$, which is consequently corrected.

Consequently, for $\omega_{est} \cong \omega_n$ the (5) is in [7,8] approximated by

$$\Delta\varphi = G_\alpha(j\omega_{est}) \tag{5}$$

$$\Delta\varphi = -\frac{\omega_{est}^2 - \omega_n^2}{K_s \omega_{est} \omega_n} \tag{6}$$

Finally, the resulting FFPLL1 phase angle estimate is equal to $\theta_{ffpll} = \theta_{est} - \Delta\varphi$. However, the shortcoming of (6) is that it is not accurate for $\omega_{est}$, which differs significantly from $\omega_n$.

In order to avoid the aforementioned phase angle compensation (6), the FFPLL2 [8] structure is proposed, outlined in Figure 3, which according to [8] statically and dynamically corresponds to the derivative element (DE) based single-phase PLL [13].

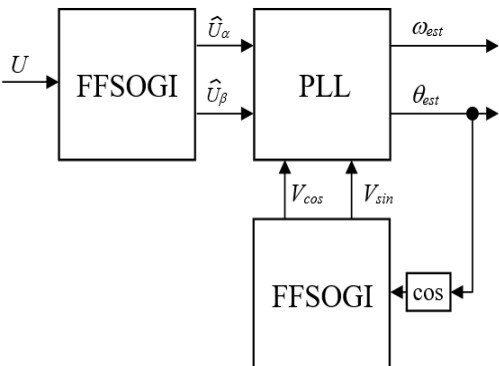

**Figure 3.** The FFPLL2 structure.

In Figure 3, $V_{cos}$ and $V_{sin}$ represent unity orthogonal signals, generated by the FFSOGI for the input signal $\cos(\theta_{est})$. However, although FFPLL2 in Figure 3 generates accurate frequency and phase angle estimates for $\omega_{est} \neq \omega_n$, it is more complex to implement.

In the next section, the improved FFPLL1 structure is proposed.

## 3. Improved FFPLL1 Structure

In this section, the improved FFPLL1 structure is proposed, which is based on the accurate phase estimation error correction. Also, two additional FFPLL structures are proposed–the first with the input signal DC offset compensation and the second with the separation of the positive sequence component in the FFPLL input.

### 3.1. Improved Correction of the Phase Angle Estimated by FFPLL1

In order to enable accurately estimated phase angle correction instead of (6), the exact correction factor is calculated based on (5). Namely, from (6) the following correction (7) of the phase angle estimated by FFPPL1 is proposed.

$$\Delta\varphi_2 = \frac{\pi}{2} - \text{atan}\left(\frac{K_s\omega_{est}\omega_n}{\omega_{est}^2 - \omega_n^2}\right) \tag{7}$$

However, it may be argued that the estimated phase angle correction (7) is more complex to implement on the contemporary microcontroller and DSP platforms compared with (6). Nevertheless, modern general purpose and specialized floating-point microcontrollers and DSP control platforms enable fast implementation of (7) in real time, which enables Equation (7) to be easily employed in different PLL applications. Consequently, the resulting modified FFPLL1 structure is outlined in Figure 4a.

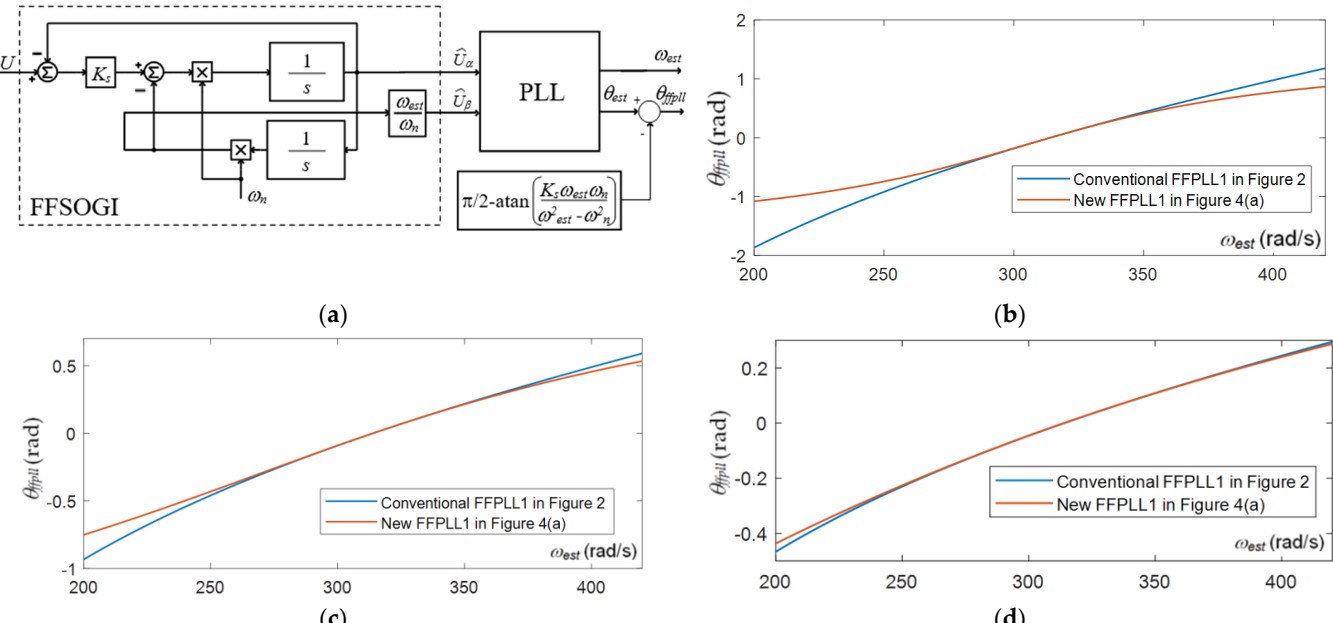

**Figure 4.** The (**a**) FFPLL1 structure with improved estimated phase angle correction, (**b**) comparison between the conventional (5) and new (7) FFPPL phase compensation, calculated for $\omega_{est} = 314$ rad/s and (**b**) $K_s = 0.5$, (**c**) $K_s = 1$, (**d**) and $K_s = 2$.

In Figure 4b–d, the comparison is outlined between the phase compensation factors in conventional FFPLL solutions (5) and proposed compensation factor (7), calculated for $\omega_{est} = 314$ rad/s and $K_s = 0.5$ in Figure 4b, $K_s = 1$ in Figure 4c, and $K_s = 2$ in Figure 4d (which are values typically used in SOGI based PLL design). It shows that the FFPLL1 with new phase compensation (7) enables accurate FFPLL operation for any FFSOGI fixed

resonant frequency value $\omega_n$, while the conventional FFPLL with approximated phase compensation (5) operates accurately only when the FFPLL input signal frequency (i.e., the estimated frequency value $\omega_{est}$ is equal to $\omega_n$) with the estimated phase angle error increasing with the increase of the difference between $\omega_n$ and $\omega_{est}$.

In the following subsection, the FFPLL parameter tuning procedure is outlined.

### 3.2. FFPLL1 Parameter Tuning Procedure

In order to propose the FFPLL1 parameter design procedure, the corresponding small-signal model needs to be devised. Consequently, in [7] the model is developed, which is presented in Figure 5.

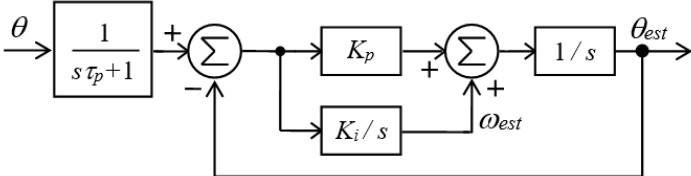

**Figure 5.** The small-signal model of FFPLL1.

In Figure 5, $\theta$ represents the FFPPL1 input signal phase angle, while $\tau_d$ represents the equivalent FFSOGI time constant $\tau_d = 2/(K_s\omega_n)$. Based on the diagram in Figure 5, the following FFPLL1 closed-loop transfer function $G_{pll}(s)$ (8) can be derived.

$$G_{pll}(s) = \frac{1}{s\tau_d + 1}\frac{K_p s + K_i}{s^2 + K_p s + K_i} \tag{8}$$

In (8), the $G_{pll}(s)$ pole $p_1 = -1/\tau_d$ depends on the FFSOGI parameters ($\tau_d = 2/(K_s\omega_n)$), while the remaining two poles can be tuned by $K_p$ and $K_i$. If the FFPLL1 is designed with poles $p_1 = -1/\tau_d$, and $p_{2,3} = -a_{pll}$, the following PLL parameter values (9) and (10) are obtained. The FFSOGI parameter $K_s$ is

$$K_i = a_{pll}^2 \tag{9}$$

$$K_p = 2a_{pll} \tag{10}$$

Furthermore, based on the analysis provided in [7,8], it can be concluded that much faster PLL dynamic performance can be achieved by FFPLL than by the PLL based on the variable frequency SOGI, based on the comparison of their corresponding small-signal models.

In the following subsection, the modification of FFPLL1 is presented, which enables the FFPLL operation with the DC offset present in the PLL input signal.

### 3.3. The FFPLL Modification with the DC Offset Compensation

In order to enable variable frequency SOGI operation with the DC offset present at the input, the SOGI with integrator (ISOGI) structure, accompanied by the corresponding parameter tuning procedure, is presented in [2], and in the following Figure 6.

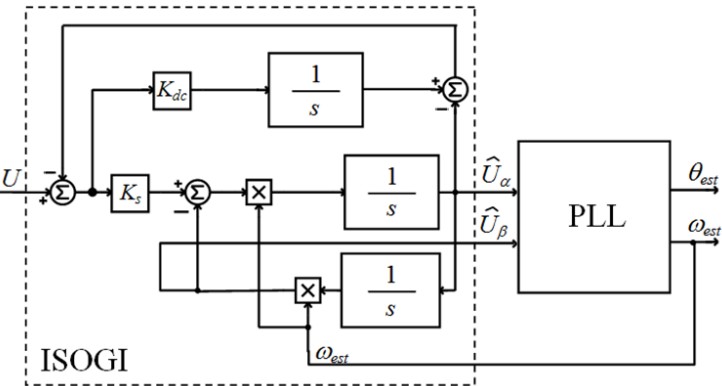

**Figure 6.** The PLL with the ISOGI orthogonal signal generator.

Based on Figure 6, the following transfer functions (11)–(12) are derived from the ISOGI input to the corresponding outputs.

$$G_{\alpha 2}(s) = \frac{\hat{U}_\alpha(s)}{U(s)} = \frac{K_s \omega_{est} s^2}{s^3 + (K_{dc} + K_s)\omega_{est} s^2 + \omega_{est}^2 s + K_{dc}\omega_{est}^3} \tag{11}$$

$$G_{\beta 2}(s) = \frac{\hat{U}_\beta(s)}{U(s)} = \frac{K_s \omega_{est}^2 s}{s^3 + (K_{dc} + K_s)\omega_{est} s^2 + \omega_{est}^2 s + K_{dc}\omega_{est}^3} \tag{12}$$

If the fixed frequency $\omega_n$ is to be employed in ISOGI (FFISOGI), the following FFISOGI equations are obtained.

$$G_{\alpha 3}(s) = \frac{\hat{U}_\alpha(s)}{U(s)} = \frac{K_s \omega_n s^2}{s^3 + (K_{dc} + K_s)\omega_n s^2 + \omega_n^2 s + K_{dc}\omega_n^3} \tag{13}$$

$$G_{\beta 3}(s) = \frac{\hat{U}_\beta(s)}{U(s)} = \frac{K_s \omega_n^2 s}{s^3 + (K_{dc} + K_s)\omega_n s^2 + \omega_n^2 s + K_{dc}\omega_n^3} \tag{14}$$

From (13) and (14), it can be concluded that $\hat{U}_\alpha$ and $\hat{U}_\beta$ are mutually orthogonal, which enables them to be used in a PLL for the input signal $U$ frequency and phase estimation. However, based also on (13) and (14), it can be concluded that the amplitude of $\hat{U}_\beta(s)$ needs to be corrected by the factor $\omega_{est}/\omega_n$, as, also, the phase error $\Delta\varphi_3 = \angle G_{\alpha 3}(j\omega_{est})$ (15) needs to be compensated, which is introduced by both (13) and (14), because $U(s)$ and $\hat{U}_\alpha(s)$ are not in phase.

$$\Delta\varphi_3 = \pi - \text{atan}\left[\frac{\omega_{est}\omega_n^2 - \omega_{est}^3}{K_{dc}\omega_n^3 - (K_{dc} + K_s)\omega_n\omega_{est}^2}\right] \tag{15}$$

Consequently, the following FFPLL1 with DC offset compensation is proposed in Figure 7.

Regarding the small-signal modeling and parameter tuning procedure of the FFPLL1 in Figure 7, it is based on the analysis outlined in [14]. Namely, it shows that the modified SOGI with DC offset compensation from Figure 6 has the same small-signal model as the conventional SOGI from Figure 1a (the first order single pole transfer function with time constant $\tau_p$ in Figure 5), and that in both cases SOGI parameters can be tuned a way that results with the same small-signal model time constant $\tau_p$ values.

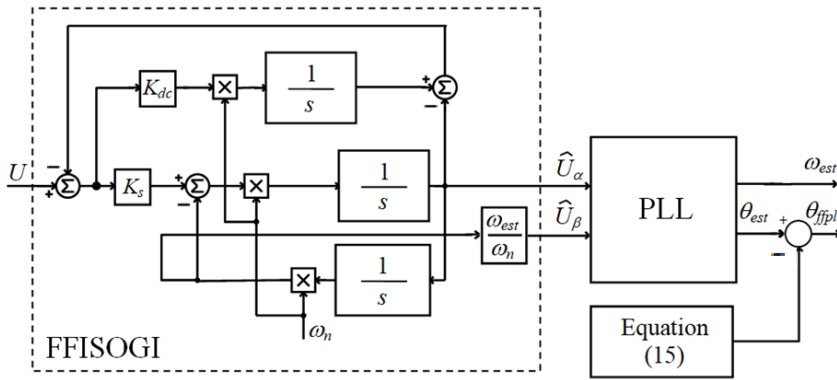

**Figure 7.** The improved FFPLL1 with the DC offset compensation.

Finally, since in both SOGI cases outlined in Figures 1a and 5 the resulting small-signal models are the same we concluded that the same small-signal models can be obtained for both FFPLL cases—the first in Figure 4 that is based on conventional SOGI, and the second in Figure 7 that is based on the SOGI with DC offset compensation. Consequently, in both cases, the same small-signal model outlined in Figure 5 is used, together with the corresponding FFPLL1 parameter tuning procedure outlined in Section 3.2.

In the following subsection, the FFPLL is presented based on the positive sequence separation from the PLL input signals.

### 3.4. The FFPLL Based Synchronization with the Positive Sequence

Because FFSOGI generates mutually orthogonal signals, substantiated by Equations (3) and (4), it can be concluded that a similar structure can be used for the positive sequence separation based on FFPLL as in the conventional variable frequency SOGI based PLL structures [14].

Consequently, the corresponding PLL structure is presented in Figure 8, where $U_\alpha$ and $U_\beta$ represent $\alpha$ and $\beta$ PLL input signal components, $\hat{U}'_\alpha$, $\hat{U}''_\alpha$, $\hat{U}'_\beta$, and $\hat{U}''_\beta$ auxiliary PLL signals, $\hat{U}^+_\alpha$ and $\hat{U}^+_\beta$ estimated positive sequence components, while the FFSOGI structure employed is as already outlined in Figure 4.

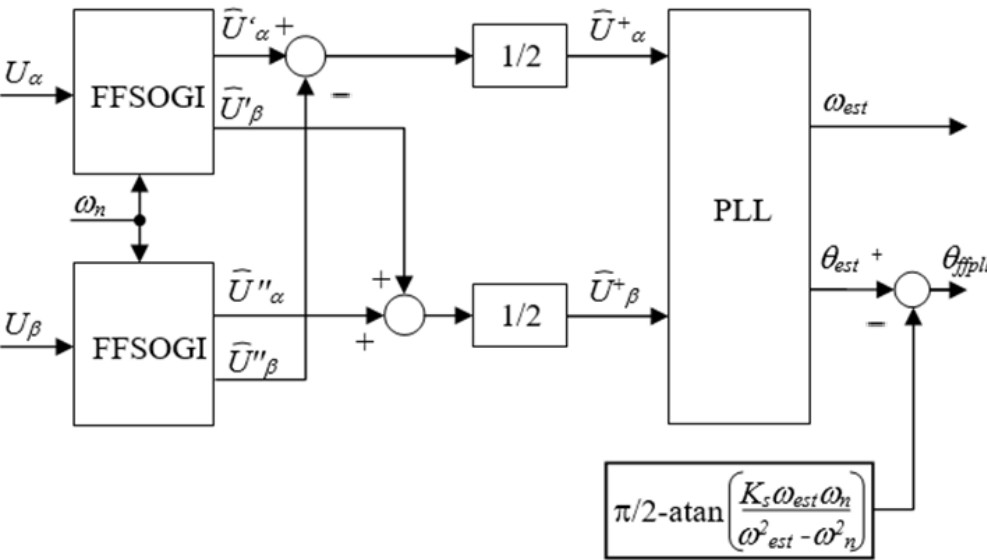

**Figure 8.** The positive sequence separation based FFPLL.

## 4. Results of Simulation Runs

In this section, the results of simulation runs are presented for the improved FFPLL1 structure and for FFPLL1 application in a PLL structure with input DC offset compensation, as well as for the case of the FFPLL application in the PLL synchronization with the positive sequence component of the input signal.

### 4.1. Simulation of the Improved FFPLL1

In this subsection, simulation results of the PLL structure from Figure 4 are presented for three different FFSOGI $K_s$ parameter values. Namely, in [1] the relation between the $K_s$ value and the equivalent SOGI time constant $\tau_d = 2/(K_s\omega_n)$ in (8) is established, with the value $K_s = 2$ recommended for practical SOGI application. However, based on Equations (8)–(10), it can be concluded that FFPLL1 PI controller parameter values $K_p$ and $K_i$ can be chosen independently from $K_s$ by means of (9)-(10) and by using the designated bandwidth value $a_{pll}$ of the closed-loop section of the FFPLL1. Consequently, the following three $a_{pll}$ values are chosen, with the corresponding $K_p$ and $K_i$ values calculated by using (9) and (10), for $K_s = 2$: (i) $a_{pll} = \omega_n = 314$ rad/s with $K_i = 98 \times 10^3$ and $K_p = 628$, (ii) $a_{pll} = 2\omega_n = 628$ rad/s with $K_i = 394 \times 10^3$ and $K_p = 1256$, and (iii) $a_{pll} = 3\omega_n = 942$ rad/s with $K_i = 887 \times 10^3$ and $K_p = 1884$.

By analyzing the simulation results in Figure 9, it can be concluded that for step input signal frequency variation $\Delta\omega_e = 31.4$ rad/s, the improved FFPLL1 successfully estimates the input signal frequency in Figure 9a, while the accurate phase angle error compensation is performed by (7), which is illustrated in Figure 9b. Also, based on the estimated frequency and phase responses, it can be concluded that their settling times correspond to the designated bandwidth frequency of FFPLL1 in Figure 5 (for $\tau_d = 2/(K_s\omega_n)$ = 1/314 s = 3.1 ms, and for three corresponding $a_{pll}$ values). Furthermore, in all three cases, the FFPLL1 settling times (between 12 and 20 ms) are significantly faster compared with the PLL with frequency adaptive SOGI [2], which commonly has settling times in the range from 40 to 50 ms for $\omega_n = 2\pi50$ rad/s.

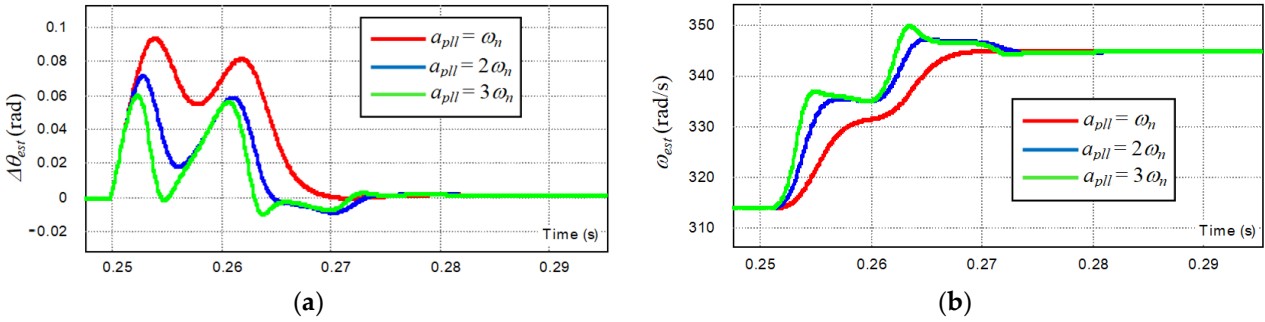

**Figure 9.** The estimated phase angle error (**a**) $\Delta\theta_{est}$ and estimated frequency (**b**) $\omega_{est}$ for three different $a_{pll}$ values, for step input signal frequency $\omega_e$ variation.

In Figure 10, improved FFPLL1 estimated phase angle and frequency values are presented for the step variation $\Delta\theta_e = 0.5$ rad of the input signal phase angle value. The simulation results show that similar response times are achieved to the results presented in Figure 9, which correspond to the three different PLL $a_{pll}$ parameter values.

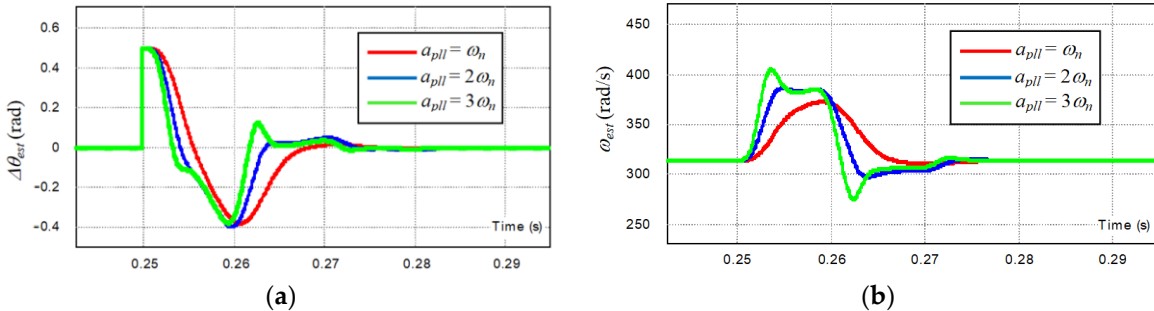

**Figure 10.** The estimated phase angle error (**a**) $\Delta\theta_{est}$ and estimated frequency (**b**) $\omega_{est}$ for three different $a_{pll}$ values, for step input signal phase angle variation.

### 4.2. Simulation of the Improved FFPLL1 with DC Offset Compensation

In this subsection, simulation results are presented of the FFPLL1 with the DC offset compensation outlined in Figure 7. Firstly, it is necessary to determine the FFISOGI parameter values $K_s$ and $K_{dc}$. This is performed by [15] with the following values proposed for FFISOGI parameters: $K_s = 1$ and $K_{dc}\omega_n = 85$, for $\omega_n = 314$ rad/s, resulting in $K_{dc} = 0.27$.

In Figure 11, the simulation results of an FFPLL with the input signal DC offset compensation are presented, for the input signal frequency step change $\Delta\omega_e = 31.4$ rad/s. By analyzing the estimated frequency and phase angle responses, the following conclusions can be drawn: (i) that the FFPLL settling times that are in the range $t_{set} = 30$–37 ms correspond to designated $a_{pll}$ and to FFISOGI dynamics (which based on [15] has the time constant $\tau_{ISOGI} = 8$–10 ms for $K_s = 1$, and $K_{DC} = 0.27$), and (ii) that the estimated phase angle correction (15) works correctly.

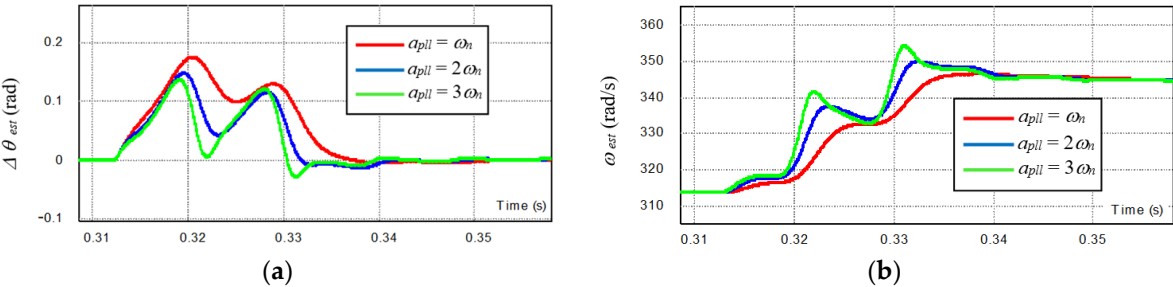

**Figure 11.** The estimated phase angle error (**a**) $\Delta\theta_{est}$ and estimated frequency (**b**) $\omega_{est}$ for three different $a_{pll}$ values, for step input signal frequency variation.

In Figure 12, the FFPLL simulated responses are presented for the DC offset equal to 0.5 V introduced at the PLL input. The simulation results show that a similar frequency and phase angle estimation times are achieved as in Figure 11, and that FFPLL with FFISOGI successfully compensates the DC offset while accurately estimating the input signal frequency and phase angle values.

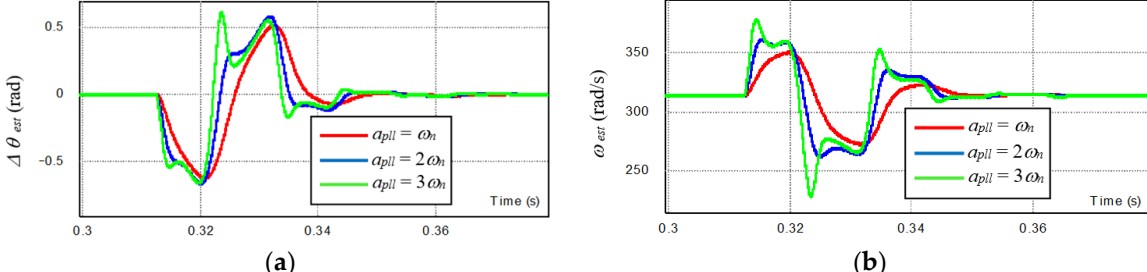

**Figure 12.** The estimated phase angle error (**a**) $\Delta\theta_{est}$ and estimated frequency (**b**) $\omega_{est}$ for three different $a_{pll}$ values, for the DC offset introduced at the FFPLL input.

### 4.3. Simulation of the Improved FFPLL with the Input Signal Positive Sequence Separation

In this subsection simulation results of the FFPLL with the positive sequence outlined in Figure 8 are presented, with the asymmetrical input signals. This is achieved by introducing the sinusoidal signal with the frequency $\omega_e$ at the input $U_\alpha$ and zero signal at the input $U_\beta$.

The simulation tests with the results outlined in Figure 13 are performed for the same set of FFSOGI and PLL parameter values as in the simulation runs in Section 4.1. By analyzing the results in Figure 13 it can be concluded that the FFPLL in Figure 8 successfully separates the positive sequence component from the input signals and estimates its frequency and phase angle with the settling times in the range $t_{set} = 10$–15 ms, which are similar to the results achieved in Section 4.1 (which should be expected since the structures in Figures 4 and 8 have the same dynamic characteristics).

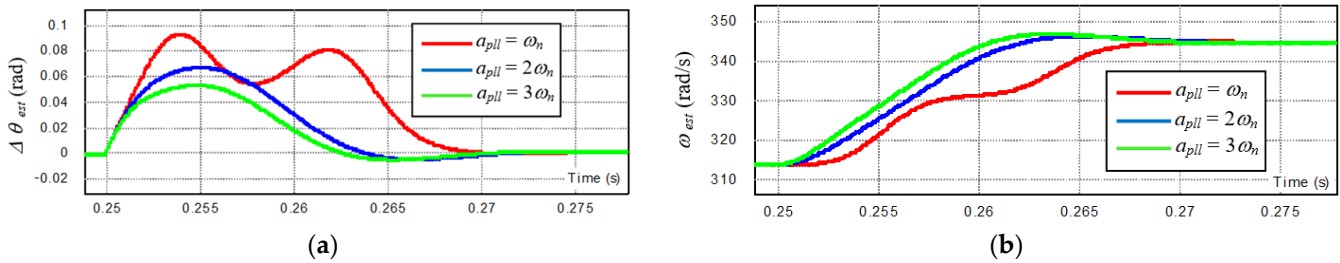

**Figure 13.** The estimated phase angle error (**a**) $\Delta\theta_{est}$ and estimated frequency (**b**) $\omega_{est}$ for three different $a_{pll}$ values, for the FFPLL with the positive sequence separation.

In the following section, the results of a wide range of experimental tests are presented and analyzed.

## 5. Experimental Tests

In this section the results of experimental tests are presented, obtained by an experimental setup comprising components outlined in Figure 14 below.

The experimental setup comprises the programmable signal generator used to emulate the grid voltage signal, a floating-point digital signal processor (DSP) TMS320F28335 based control card used to implement the FFPLL, and personal computer (PC) running software for real-time signal acquisition and FFPLL parameter settings.

The following subsections present the experimental results of three different FFPLL applications.

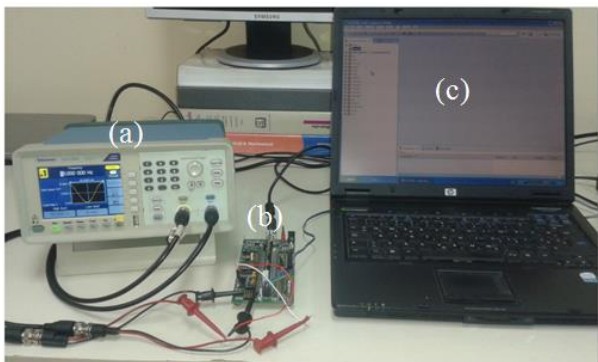

**Figure 14.** Experimental setup, (**a**) programmable signal generator, (**b**) TMS320F28335 microcontroller based PLL platform, (**c**) PC used for the experimental data acquisition.

*5.1. Experimental Tests of the FFPLL1 in Figure 4*

This subsection details the experimental tests of the FFPLL structure in Figure 4, for the three sets of parameters $a_{pll}$ calculated in Section 4.1, $K_s$ = 2, and nominal frequency value $\omega_n$ = 314 rad/s outlined in Section 4.1.

By analyzing the results of the experimental tests presented in Figure 15 (for 31 rad/s step frequency variation in (a), and 0.5 rad step phase angle change in (b)), it can be concluded that the estimated frequency settling times in the range $t_{set}$ = 15–20 ms are achieved, which are similar to the simulation results in Section 4.1 for the same FFPLL1 structure in Figure 4.

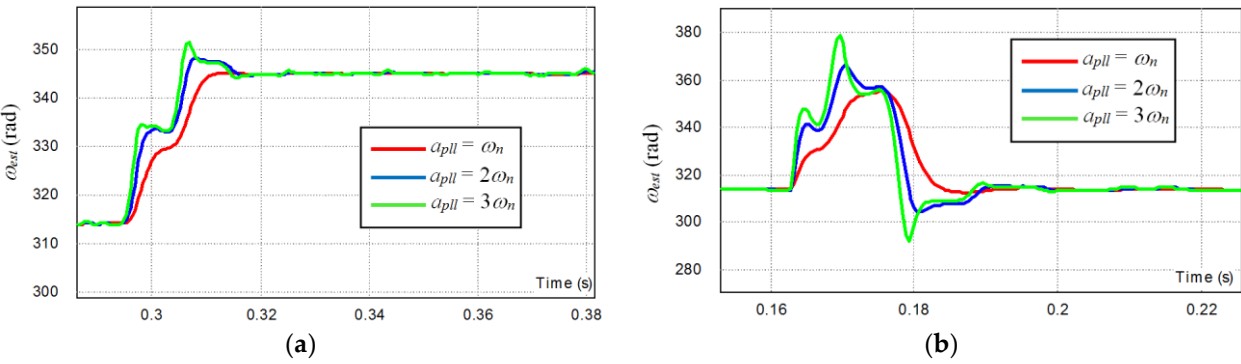

**Figure 15.** The estimated frequency for (**a**) input signal frequency step variation 31 rad/s, and for (**b**) input signal step phase angle variation 0.5 rad, for the FFPLL in Figure 4, for three different values of PLL parameters $a_{pll}$ and $K_s$ = 2, for nominal FFPLL frequency $\omega_n$ = 314 rad/s and for input signal frequency $\omega_e$ = 314 rad/s.

*5.2. Experimental Tests of the FFPLL1 in Figure 7*

In this subsection the experimental tests of the FFPLL1 structure in Figure 7 are presented, for three different $a_{pll}$ values, $K_s$ = 1, $K_{DC}$ = 0.27, and for nominal frequency $\omega_n$ = 314 rad/s.

In Figure 16 the results of experimental tests are presented comprising the responses for a 31 rad/s step frequency change in (a), 0.5 rad step phase angle change in (b), and a DC offset introduced at the FFPLL input in (c). In all three cases, estimated frequency settling times in the range $t_{set}$ = 30–35 ms are achieved, which corresponds to the results in Section 4.2, which include the simulation of the same FFPLL structure in Figure 7 for the same set of parameters.

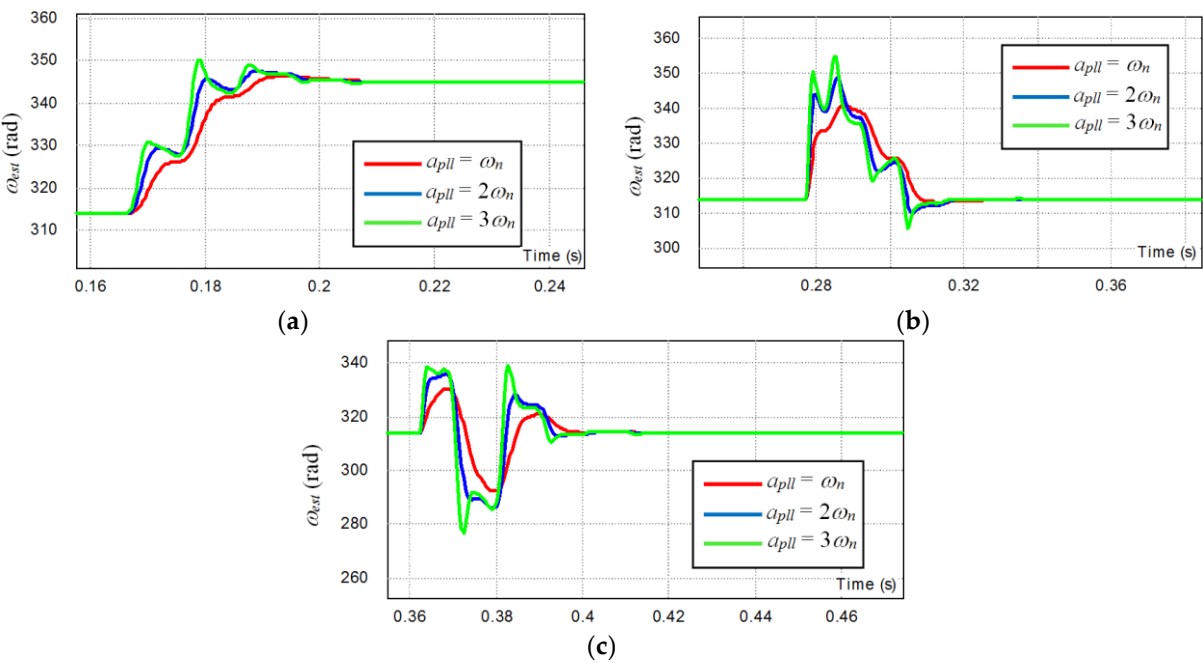

**Figure 16.** The estimated frequency for (**a**) input signal frequency step variation 31 rad/s, and for (**b**) input signal step phase angle variation 0.5 rad, and (**c**) for DC offset introduced at the FFPLL input, for the FFPLL in Figure 7, for three sets of PLL parameters $a_{pll}$, nominal FFPLL frequency $\omega_n$ = 314 rad/s and for input signal frequency $\omega_e$ = 314 rad/s.

### 5.3. Experimental Tests of the FFPLL1 in Figure 8

In this subsection, experimental results for the FFPLL structure in Figure 8 are presented, for three different $a_{pll}$ values calculated in Section 4.1, and for $K_s$ = 2.

By analyzing the experimental results in Figure 17 (obtained for 31 rad/s input frequency step variation in (a), for 0.5 rad step input phase angle variation in (b), and for the nominal frequency value $\omega_n$ = 314 rad/s), it can be concluded that in both cases (a) and (b) the estimated frequency settling times $t_{set}$ = 10–15 ms are achieved, which are similar to the corresponding simulation results in Section 4.3.

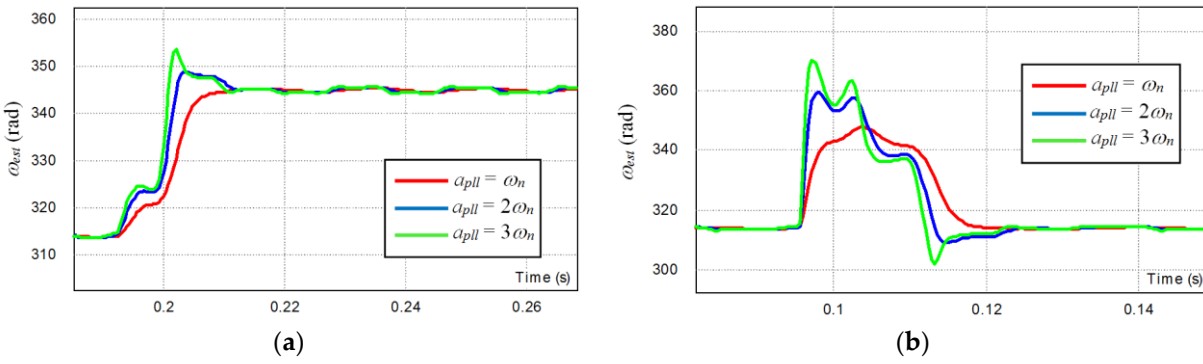

**Figure 17.** The estimated frequency for (**a**) input signal frequency step variation 31 rad/s, and for (**b**) input signal step phase angle variation 0.5 rad, for the FFPLL in Figure 8, for three different values of PLL parameters $a_{pll}$ and $K_s$ = 2, for nominal FFPLL frequency $\omega_n$ = 314 rad/s and for input signal frequency $\omega_e$ = 314 rad/s.

## 6. Conclusions

In this paper, an improved fixed frequency PLL is presented, with the proposed contribution comprising the modified algorithm for the compensation of the estimated phase angle value, which is typically required by an FFPLL. The analysis outlined in

the paper shows that the novel FFPLL enables accurate phase angle estimation for any combination of the fixed FFSOGI resonant frequency value $\omega_n$ and estimated input signal frequency $\omega_{est}$, while the conventional FFSOGI operates accurately only for $\omega_n = \omega_{est}$, with the phase angle estimation error increasing with the difference between $\omega_n$ and $\omega_{est}$. Also, the corresponding FFPLL parameter tuning procedure is proposed and tested by a series of simulation and experimental tests. Three different FFPLL applications are examined: (i) single-phase PLL with no DC offset at the input, (ii) single-phase PLL designed to compensate a DC offset at the input, and (iii) PLL designed to separate a positive sequence component and to estimate its phase angle and frequency. For all three cases, corresponding simulation and experimental tests were performed, for three sets of FFPPL parameters, and for step variations of the input signal frequency, phase angle, and a DC offset. Both simulation and experimental tests verified the proposed method's dynamic and static performance, with improved dynamic performance compared to the conventional adaptive filter based single-phase PLL applications. In a global context, by the method outlined in the paper, the existing frequency non-adaptive FFPLL algorithms (which are dynamically superior to conventional frequency adaptive PLL solutions) were further improved, by enabling analytically accurate phase angle estimation in the simplest FFPLL1 case.

**Author Contributions:** Conceptualization, D.S.; Formal analysis, D.S. and T.T.; Investigation, D.S.; Methodology, D.S.; Software, D.S.; Supervision, L.M.G.; Validation, D.S., T.T. and L.J.N.; Writing – original draft, D.S.; Writing – review & editing, D.S., T.T. and L.M.G. All authors have read and agreed to the published version of the manuscript.

**Funding:** This research received no external funding.

**Data Availability Statement:** The data presented in this study are available on request from the corresponding author.

**Conflicts of Interest:** The authors declare no conflict of interest.

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
