# Peer review of "Improved Fixed-Frequency SOGI Based Single-Phase PLL"

_energies, doi:10.3390/en15197297_

Round 1

Reviewer 1 Report

In this work, the authors propose an improved single-phase second-order generalized integrator (SOGI) fixed frequency phase-locked loop (FFPLL). On the one hand, this paper modify the estimated PLL input signal angle correction factor, FFPLL with DC offset and corresponding angle estimation correction techniques are proposed. On the other hand, based on the new FFPLL structure, the PLL with positive order separation is summarized. Moreover, through simulation and experiment, the proposed technology is analyzed and verified, which proves the accuracy and effectiveness of the proposed phase-locked loop technology.

In my opinion, this work is interesting and the arguments are convincing, which may arouse extensive research interest in the communication community. I would like to recommend the publication of this manuscript in Energies.

----Minor corrections and suggestions are listed below:

1). The authors can adjust the structure of the paper to make it more compact, so as to better guide the readers.

2). Some grammar and writing errors should be corrected in the revised paper.

3). Can the FFPLL proposed in this paper be used to construct microwave antennas with specific phase? Which may provide a good way to realize the novel near-field photonic routing with meat-sources [Science 348(6242), 1448-1451 (2015); Adv. Photon. 3(3), 036001 (2021)] and promote the development of spin photonics.

Author Response

Question 1

1) The authors can adjust the structure of the paper to make it more compact, so as to better guide the readers.

Answer 1

Additional text was added in the revised version of the paper to better guide the readers, including the structural changes, especially in the introduction section.

Question 2

2) Some grammar and writing errors should be corrected in the revised paper.

Answer 2

Detected grammar and writing errors were corrected in the revised version of the paper.

Question 3

3). Can the FFPLL proposed in this paper be used to construct microwave antennas with specific phase? Which may provide a good way to realize the novel near-field photonic routing with meat-sources  [Science 348(6242), 1448-1451 (2015); Adv. Photon. 3(3), 036001 (2021)] and promote the development of spin photonics.

Answer 3

The limitation of the proposed algorithm applications is determined by the control platform that it can be realized on. Namely, in order to process the input FFPLL signal it is necessary to provide the control platform that enables digital signal processing with the sampling and actuation frequency at least about 10 to 12 times higher than the frequency of the FFPLL input.

Consequently, the application of the proposed FFPLL algorithm in applications outlined in papers [Science 348(6242), 1448-1451 (2015); Adv. Photon. 3(3), 036001 (2021)] would require digital signal processing (signal sampling and algorithm calculations) in GHz frequency ranges, which can currently be provided, for example, by the Xilinx RFSoC platforms. However, there is also a possibility of the analogue FFPLL applications, where one of the proposed solutions is outlined in [R1].

[R1] Breems, L. J., van Sinderen, J., Fric, T., Stoffels, H., Fritschij, F., Brekelmans, H., ... & Lassche, G. (2019, June). A Full-Band Multi-Standard Global Analog & Digital Car Radio SoC with a Single Fixed-Frequency PLL. In 2019 IEEE Radio Frequency Integrated Circuits Symposium (RFIC) (pp. 315-318). IEEE.

In order to point out the analysis outlined in this answer to Question 3, the following paragraph was added into the introduction section of the paper, together with the reference [R1]:

“Regarding the application of the FFPLL algorithms in radio frequency (RF) and micro-wave applications, it is limited by the features of the employed control platform. Namely, in order to implement an FFPLL based on a digital signal processing (DSP) in an RF application a specialized RFSoC platform [11] could be employed. However, there is a possibility of an analogue FFPLL application [12], which can overcome some of the shortcomings of a DSP based solution.”

Reviewer 2 Report

energies-1929155

In this paper an improved single-phase second order generalized integrator (SOGI) fixed-frequency phase-locked loop (FFPLL) is presented. The proposed improvement consists of the modification of the estimated PLL input signal angle correction factor, which is in this paper calculated and implemented with the exactly accurate value, while in the existing literature the approximated correction value is employed. Also, in this paper the FFPLL with DC offset is presented, together with the corresponding estimated angle correction technique.

However, novelties are limited and I personally doubt its value for practical applications. Please find the technical comments:

·         several fixed-frequency phase-locked loop is developed for real time application. What is new in this? What modifications authors have done. It is not clear.

·         What message does Fig. 1 convey? Conventional PLL structure and its modelling is already published in several article. why the same is repeated here?

·         Same as previous comment for Fig. 2, Fig. 3 etc…

·         Stability analysis is required for the block diagram given in Fig. 7. Additional DC offset compensation may create the stability issue in the system.

·         It is written that system is validated in hardware, but I don’t find any hardware photo or results in the manuscript.

·         Kindly include the comparison of time domain specification of the system and comment on the results with the obtained.

·         Without comparison, work cannot be claimed as superior to the other.  

Please find some general comment to enhance the structure of the paper:

  1. Clarify better the innovation of this work in the abstract and in the main text.
  2. The introduction has been vaguely written. My suggestion is to divide the introduction into three subsections: 1) motivation and incitement, 2) literature review and 3) contribution and paper organization.
  3. A short paragraph introducing the problem statement should be included at the end of the Introduction section. What is the central problem in a sentence that manuscript wants to solve?
  4. The main contribution of the paper should be highlighted and emphasized. It would be great if the drawbacks and gaps of literature are clear and, particularly, how the proposed approach aims at filling these gaps.
  5. Extend the conclusion with more general usability. What are the benefits of the results in a global context? Please explain this better in the manuscript.
  6. There are some grammatical errors and typos that should be corrected before publication.
  7. Include the latest reference.

Author Response

Question 1

Several fixed-frequency phase-locked loop is developed for real time application. What is new in this? What modifications authors have done. It is not clear.

Answer 1

The main contribution proposed in the paper comprises the modification of the conventional FFPLL structure, which is outlined in the Figure 2. Namely, the application of conventional FFPLL comprises a feedforward compensation defined by equation (5) of the estimated phase angle qest, in order to obtain the resulting phase angle qffpll. However, the compensation factor (5) represents an approximation of the accurate compensation value (7), calculated for the input signal frequency wn@west. Consequently, the contribution proposed in this paper comprises the new accurate FFPLL estimated phase angle compensation factor (7), instead of the approximated compensation factor (5) that is applied in conventional FFPLL applications.

In order to point this out, following text was added into the introduction section of the paper:

“Namely, in conventional FFPLL applications [7] phase compensation value is approximated for the estimated input signal frequency value close to fixed resonant frequency of the employed SOGI term, while the new proposed solution comprises an analytically calculated accurate phase compensation factor. In this way, accurate operation of the FFPLL for the much wider differences between the input signal frequency and SOGI term fixed resonant frequency value, which was not the case in conventional FFPLL solutions.”

In the main body of the paper, following text was also introduced, for the same reason:

“In figures 4(b)-4(d), the comparison is outlined between the phase compensation factors in conventional FFPLL solutions (5) and proposed compensation factor (7), calculated for west=314 rad/s and Ks=0.5 in 4(b), Ks=1 in 4(c), and Ks=2 in 4(d) (which are values typically used in SOGI based PLL design). It shows that the FFPLL1 with new phase compensation (7) enables accurate FFPLL operation for any FFSOGI fixed resonant frequency value wn, while the conventional FFPLL with approximated phase compensation (5) operates accurately only when the FFPLL input signal frequency (i.e., the estimated frequency value west is equal to wn) with the estimated phase angle error increasing with the increase of the difference between wn and west.”

The main contribution was also emphasized in the conclusion section of the paper:

“The analysis outlined in the paper shows that the novel FFPLL enables accurate phase angle estimation for any combination of the fixed FFSOGI resonant frequency value wn and estimated input signal frequency west, while the conventional FFSOGI operates accurately only for wn=west, with the phase angle estimation error increasing with the difference between wn and west.”

Question 2

What message does Fig. 1 convey? Conventional PLL structure and its modelling is already published in several article. why the same is repeated here?

Answer 2

The Figure 1(a) comprising the conventional frequency adaptive PLL based on SOGI term, combined with equations (1) and (2), is used to explain the main working principle of the FFPLL that is derived from the structure that it illustrates. This is emphasized by adding the following sentence prior to equations (1) and (2):

“Namely, equations (1) and (2) represent the basis for the derivation of the FFPLL operating equations, which are outlined in the following part of the paper.”

The Figure 1(b) shows the conventional outline of the PLL based on the employed orthogonal signal generation algorithm, which is useful since corresponding PLLs based on the FFSOGI are used throughout the paper.

The Figure 2 is introduced in the paper, since it illustrates the conventional FFPLL structure with the approximated estimated phase angle compensation, which represents the basis for the implementation of the novel FFPLL with the accurate estimated phase angle compensation, which is proposed in this paper.

The Figure 3 is interesting because it illustrates the FFPLL structure that, actually, does not have a problem with the estimated phase angle error caused by the difference between wn and west. However, it is more complex than the solution outlined in this paper, with both yielding similar results.

Question 3

Stability analysis is required for the block diagram given in Fig. 7. Additional DC offset compensation may create the stability issue in the system.

Answer 3

The dynamic modeling of the FFPLL is outlined in [7], and it yields the small-signal model that is outlined in Figure 5. The resulting FFPLL parameter tuning procedure is outlined in [7-8] and in subsection 3.2 of this paper, and the analysis in [7-8] shows that much faster dynamic response can be achieved by FFSOGI based FFPLL than with conventional SOGI based PLL.

Regarding the influence of the modified DC-offset prune SOGI in Figure 6. on the resulting FFPLL dynamics (which was the basis of Question 3), the reasoning behind the employed FFPLL small-signal modeling and parameter tuning procedure is based on the small-signal model analysis outlined in paper [13], with the following corresponding text included in the paper:

“Regarding the small-signal modeling and parameter tuning procedure of the FFPLL1 in Figure 7, it is based on the analysis outlined in [13]. Namely, it shows that the modified SOGI with DC offset compensation from Figure 6 has the same small-signal model as the conventional SOGI from Figure 1(a) (the first order single pole transfer function with time constant tp in Figure 5), and that in both cases SOGI parameters can be tuned a way that results with the same small-signal model time constant tp values.

Finally, since in both SOGI cases outlined in Figures 1(a) and 5 the resulting small-signal models are the same we concluded that the same small-signal models can be obtained for both FFPLL cases–the first in Figure 4 that is based on conventional SOGI, and the second in Figure 7 that is based on the SOGI with DC offset compensation. Consequently, in both cases, the same small-signal model outlined in Figure 5 is used, together with the corresponding FFPLL1 parameter tuning procedure outlined in subsection 3.2.”

Question 4

It is written that system is validated in hardware, but I don’t find any hardware photo or results in the manuscript.

Answer 4

In new Figure 14, the picture of the used experimental setup is introduced in the paper.

Question 5

Kindly include the comparison of time domain specification of the system and comment on the results with the obtained.

Without comparison, work cannot be claimed as superior to the other.

Answer 5

The main contribution that is introduced in the paper does not include any dynamical improvements of the results achieved by conventional FFPLL outlined in Figure 2. Instead, the static estimation accuracy is improved by introducing the new phase angle compensation factor outlined in Figure 4 into the conventional FFPLL. In this way, FFPLL structure is provided that accurately operates for all wn and west parameter values, contrary to conventional FFPLL in Figure 2 that accurately operates only in the case when wn=west.

To illustrate this fact, additional Figure 4(b) is introduced in the paper, together with the corresponding text that was already mentioned in the Answer 1 of the Reviewer 2.

Question 6

Clarify better the innovation of this work in the abstract and in the main text.

Answer 6

The issues mentioned in this question were addressed in the Answer 1 to Reviewer 2, with the corresponding modifications introduced in the paper, as was requested.

Question 7

The introduction has been vaguely written. My suggestion is to divide the introduction into three subsections: 1) motivation and incitement, 2) literature review and 3) contribution and paper organization.

Answer 7

The new introduction section was divided in the proposed three subsections. Also, in the 1.1 Motivation subsection the following text was added:

“The main motivation behind the work outlined in this paper emerges from the possibility to improve significantly the estimation precision of existing FFPLL solutions. Also, the importance of the FFPLL solutions is based on several analyses, which show that implementation of frequency non-adaptive FFPLL solutions, when compared to conventionally used frequency adaptive PLL solutions, introduces significant improvements in resulting stability, maximum response speed, and phase-locked loop robustness of operation.”

Question 8

A short paragraph introducing the problem statement should be included at the end of the Introduction section. What is the central problem in a sentence that manuscript wants to solve?

Answer 8

In order to address this issue, the following sentence was added at the end of the introduction section:

“Consequently, the problem statement comprises an effort to improve the phase angle estimation accuracy in the complete input signal frequency range for the existing FFPLL solutions, which are based on several analyses [7-8] dynamically superior to conventionally used frequency adaptive SOGI based PLLs.”

Question 9

The main contribution of the paper should be highlighted and emphasized. It would be great if the drawbacks and gaps of literature are clear and, particularly, how the proposed approach aims at filling these gaps.

Answer 9

In order to point out the drawbacks of existing FFPPL solutions, and means to solve these issues, following text was added to subsection 1.1:

“Namely, the main drawback of existing FFPLL solutions comprises the approximated estimated phase angle compensation, contrary to the proposed solution, which is based on the analytically accurate compensation factor.”

Question 10

Extend the conclusion with more general usability. What are the benefits of the results in a global context? Please explain this better in the manuscript.

Answer 10

In order to answer this question, following text was added at the end of the conclusion section:

“In a global context, by the method outlined in the paper, the existing frequency non-adaptive FFPLL algorithms (which are dynamically superior to conventional frequency adaptive PLL solutions) are further improved, by enabling analytically accurate phase angle estimation in the simplest FFPLL1 case.”

Question 11

There are some grammatical errors and typos that should be corrected before publication.

Answer 11

Detected grammatical and typing errors were corrected throughout the paper.

Question 12

Include the latest reference.

Answer 12

Three up-to-date references [11]-[13] were added in the revised version of the paper.

Round 2

Reviewer 2 Report

Now the paper is well revised. It can be accepted in its current form.